# Resveratrol: Its Path from Isolation to Therapeutic Action in Eye Diseases

**DOI:** 10.3390/antiox11122447

**Published:** 2022-12-12

**Authors:** Roxana Pop, Adela Daescu, Dumitrita Rugina, Adela Pintea

**Affiliations:** Faculty of Veterinary Medicine, University of Agricultural Sciences and Veterinary Medicine, Mănăştur 3-5, 400372 Cluj-Napoca, Romania

**Keywords:** resveratrol, delivery systems, ocular pathologies, VEGF, sirtuins

## Abstract

Due to the confirmed therapeutic potential of resveratrol (Rv) for eye diseases, namely its powerful anti-angiogenic and antioxidant effects, this molecule must be studied more deeply. Nowadays, the pharmaceutic and pharmacokinetic available studies offer a troubling picture because of its low stability and bioavailability. To overcome this problem, researchers started to design and create different delivery systems that could improve the delivery amount of Rv. Therefore, this review aims to shed light on the proper and efficient techniques to isolate, purify and quantify the Rv molecule, and how this therapeutic molecule can be a part of a delivery system. The Rv great impact on aspects regarding its stability, bioavailability and absorption are also debated here, based on the existent literature on in vitro and in vivo human and animal studies. Moreover, after its absorption the Rv influence at the molecular level in ocular pathologies is described. In addition, the present review summarizes the available literature about Rv, hoping that Rv will gain more attention to investigate its unexplored side.

## 1. Introduction

The use of plant extracts has brought significant benefits to human health. The performance of specific biological activities as well as the efficiency and safety in the use of plant compounds have supported the development of effective therapies in the treatment of various diseases. Having a central role in health management, natural compounds have attracted increasing attention in areas such as nutrition, pharmacology, biotechnology or nanotechnology. Unfortunately, almost all plant-derived biomolecules have a low physiological stability, reduced bioavailability, long blood circulation time, or non-selectivity regarding the target tissue. Furthermore, the oral administration of plant-derived biomolecules proved to be efficient only if high doses are applied. Lately, much effort has been invested in finding novel technologies to provide them protection till the target site is reached, by their entrapment inside a properly engineered system.

Resveratrol (3,5,4′-trihydroxystilbene, Rv) could be one of the future therapeutic agents for eye diseases, because it is a hydrophilic small molecule (228 Daltons) able to enter into the retina cells, downregulating the vascular endothelial growth factor (VEGF)—an angiogenic factor—and improving the antioxidant activity of the cells. Unfortunately, the therapeutic potential of Rv is difficult to exploit due to its low bioavailability and stability. Additionally, the beneficial effects of Rv were revealed mostly in cell cultures and animal models, while relatively few studies were performed on human subjects. To overcome these disadvantages, various resveratrol delivery systems have been developed that have proven to be promising in terms of eye diseases treatment.

In view of the background presented above, the aim of the present work is twofold: to bring together the essential and relevant information about resveratrol and to build, based on them, new research directions to fill the remaining gaps regarding the beneficial effect of the polyphenol in the treatment of eye diseases. The first part of this paper presents the chemical profile and characterization methods of resveratrol. The mechanisms of action of the compound in eye disease management is also discussed. Finally, a description of the delivery systems developed so far for resveratrol is given together with highlights of their implementation, with examples in the literature.

## 2. The Occurrence and Sources of Resveratrol

The plant kingdom was the basis of medicine in ancient times. Today, the use of plants by scientists is even more extensive. For example, enzymes, regulatory proteins, primary or secondary metabolites, or plant transport molecules may be prolific sources of new drugs [1]. The main advantages in using natural molecules are the high chemical and structural diversity, the rigidity of the molecules, the large number of oxygen and carbon atoms and the multitude of hydrogen bonds—which can act as acceptors or donors [2].

Polyphenols are the largest and most diverse class of secondary metabolites produced by plants. According to the chemical structure, the four main categories of polyphenols are flavonoids, phenolic acids, stilbenes and lignans [3]. Even though the main role that these compounds play in plants is in defense, each sub-class of compounds has its own particularities in terms of distribution and functionality [4].

The representative compound for the stilbenes class is resveratrol (3,5,4′-trihydroxystilbene) (Rv). It is a natural phytoalexin produced by plants in the fight against pathogens and environmental factors that trigger stress [5]. The history of resveratrol begins with its discovery, dated in 1940. The compound was extracted for the first time from the roots of *Veratrum grandiflorum* by Japanese researchers [6]. It was later detected and isolated from a traditional Chinese medicinal plant called *Polygonum cuspidatum* [7]. However, more attention has been paid to resveratrol since 1992, when it was linked to the so-called “French paradox”. An interesting phenomenon was observed in France at that time: even though the prevalence of smoking, saturated fat intakes, blood pressure and serum cholesterol levels were the same as in other countries, mortality from ischemic heart disease was very low. In those days, the conclusion reached by scientists was that moderate consumption of red wine (two drinks/day) with a meal, a habit of the French people, brings benefits and protects against heart disease [8].

Many plants were identified as a source of Rv. So far, the main source is considered to be grapes, as suggested by the discussion on the “French paradox”. The total amount of stilbene in different red wines varies between 98 μg/100 mL to over 1803 μg/100 mL depending on the climate, grape variety and vinification approach [9]. In contrast, white wines have, on average, a six times lower concentration of *trans*-resveratrol [10]. Other natural sources are pistachios, peanuts, blueberries, raw cranberry juice, chocolate, strawberries, cocoa and red currants [11,12]. Interestingly, the *trans*-resveratrol glycoside content was analyzed in several parts of the plant popularly called Japanese knotweed or Itadori plant (*Polygonum cuspidatum* Sieb et Zucc): young stem (497 μg/g) vs. old stem (83 μg/g), young leaf (867 μg/g) vs. old leaf (370 μg/g) and commercial root (1653 μg/g). Itadori tea also showed a high content of 974 μg total resveratrol/100 mL [9]. In another experiment, a maximum amount of 18.4 ± 1.6 µg resveratrol/g dry weight was measured in the skin of the tomato fruit [13]. In peanuts, the concentration of the compound is low: 0.13 μg/g in roasted peanuts, 1.19 μg/g in peanut roots, 2.05 μg/g fresh weight in leaves and 1.7–7 μg/g in canned boiled peanuts [14]. Additionally, small amounts were identified in raw fruits: 140.0 ± 29.9 pmol/g in highbush blueberries and 71.0 ± 15.0 pmol/g in bilberries [15].

## 3. Biochemical Insight into Biosynthesis Pathways of Resveratrol and Its Derivatives

The chemical profile of resveratrol has been successfully developed by scientists through detailed investigations. At present, aspects related to the chemical structure, biosynthesis and existing derivatives are known.

With a molecular weight of 228.25 g/mol and a molecular formula of C_14_H_12_O_3_, Rv is available as a stilbene monomer [16]. With the unfolding of the molecule, a C6-C2-C6 carbon skeleton can be seen, which represents two phenyl rings linked by an ethylene bridge [17] (Figure 1). The presence of the double chemical bond gives a note of rigidity to the compound, not allowing its free rotation. Thus, there are two possible configurations: *cis* and *trans* [3]. The structure of the molecule is completed by the display of three functional groups of hydroxyl type (-OH) on the two rings, in specific places: on ring A in positions C3 and C5, and on ring B in position C4′. These positions, through the presence of functional groups, allow a wide range of changes in the structure of the molecule that give rise to derivatives with specific action. Methyl, sugar or methoxy groups may be added by substitution reactions. Thus, changes such as oligomerization, isoprenylation, glycosylation, methoxylation and isomerization make it possible for dimeric, trimeric, tetrameric and larger polymers to exist [11,17]. To date, more than 300 oligomers have been identified in plants. Their structure includes between two and eight repetitive units of Rv [18].

The path of Rv biosynthesis begins with a phenylalanine molecule which, through several enzymatic reactions, is converted to p-coumaroyl-CoA. The next step is a condensation reaction between three molecules of malonyl-CoA and one molecule of p-coumaroyl-CoA, thus forming the molecule of interest. Therefore, *trans*-resveratrol is formed by an aldol-type reaction and under the action of the key enzyme stilbene synthase (STS) [16]. The STS enzyme consists of two subunits of 40 and 45 kDa. Although its expression is constitutive, under the influence of biotic or abiotic stressors its synthesis intensifies [19]. The activation of STS genes may be different between plant species or may be tissue-specific. It is possible that the accumulation of stilbenes is done depending on the environmental factors and the stage of development of the plant [20]. The STS gene family contains an impressive number of members with different characteristics. After analyzing some tissues and organs of grapevine, 41 proteins were identified [21]. Additionally, 31 STS genes were discovered in the *V. vinifera* genome, the first two being located on chromosome 10, and the rest on chromosome 16 [22]. The results of another study indicate the presence of 40 simple sequence repeat (SSR) loci with high polymorphism in the grapevine genome, at the level of chromosome 16, suggesting that the genes located here are more involved in Rv biosynthesis than those present on other chromosomes [23]. In the grape plant, the highest amount of mRNA corresponding to STS gene expression was identified in leaves, roots and petiole. At the subcellular level, the cytoplasm, the cell wall and the vacuole were identified as the preferred location for the monitored compound [24].

The synthesis of flavonoid compounds starts from the same two precursors, malonyl-CoA and p-coumaroyl-CoA (molecular ratio 3:1). The process is catalyzed by the enzyme chalcone synthase (CHS) and leads to the formation of the first intermediate in the path of flavonoid biosynthesis, a compound called chalcone. Therefore, the two enzymes compete when the substrate molecules are available [19]. A schematic representation of the biosynthetic process can be seen in Figure 2.

Rv can also be manufactured in the laboratory. The main types of chemical reactions used are: Horner–Wadsworth–Emmons reaction, Perkin reaction, Heck reaction and Wittig reaction—each with its advantages and disadvantages [25]. On an industrial scale, it can be produced by using plant cell cultures or microorganisms (bacteria and yeast). With numerous biotechnological techniques available, Rv can be obtained quickly and at low cost [26].

Once synthesized, the Rv molecule undergoes chemical changes. These can take place in three distinct areas: the C-C double bond in the ethylene bridge, the hydroxyl groups or the benzene ring. Studies shown that these changes can make a significant difference in improving or differentiating the bioactivity of Rv [25]. Thus, the investigation of the structure–activity relationship (SAR) has attracted the attention of researchers.

A number of methoxylated analogs of Rv have shown chemo-preventive, antioxidant and anti-inflammatory activity. Compared with Rv, 3,4,4′-trimethoxystilbene and 3,4,2′,4′-tetramethoxystilbene showed higher cytotoxicity toward human promyelocytic (HL-60) and monocytic leukemia (THP-1) cells by intensifying cell apoptosis and arresting cells in the G2/M phase [27]. By applying a treatment with pterostilbene (3,5-di-methoxy-4-hydroxystilbene) on two different melanoma cell lines (MCF-7 and MDA-MB-231), the activity of caspase 3/7 and implicitly cellular apoptosis were intensified [28]. The protective effect of methoxylated analogs against DNA oxidation was evaluated in an in vitro experiment performed on CHO (Chinese hamster ovary) cell line. Pterostilbene reduced the damage caused by hydrogen peroxide by 85.5%, trimethoxystilbene by 43.7% and resveratrol by 21.1% [29].

Hydroxylated Rv analogs represent a promising class of antioxidants. The scavenging activities of the derivatives possessing hydroxyl groups in positions 3′, 4 or 5′ proved to be superior to Rv [30]. In another study, ortho-dihydroxyl or 4-hydroxy-3-methoxyl compounds were effective against chemical induced human low density lipoprotein (LDL) peroxidation [31]. Additionally, 3,4-dihydroxyl groups give resveratrol-derived compounds the ability to exert potent apoptotic activity, an effect demonstrated in vitro on HL-60 and Jurkat human leukemia cell lines [32]. Another compound called piceatannol (3,5,3′,4′-tetrahydroxy-*trans*-stilbene) was identified as an inhibitor of COX-2, an enzyme involved in the inflammatory process, but also of Nf-kB in an in vivo mouse skin study [33]. Moreover, the compound 3,3′,4′,5-tetra-*trans*-hydroxystilbene proved to be an excellent selective inhibitor of COX-2 [30].

The activity of oligomers and glycosylated compounds has also been analyzed in various experimental studies. By substituting the hydroxyl from position C3 with a glucoside group, resveratrol-3-O-beta-D-glucoside, also called piceid or polydatin, is obtained. Piceid has been extracted and isolated from many fruits and vegetables, the main source being grape skin [34]. Therefore, it is found in red, white and sparkling wines [35]. With the conformational changes that occur, the piceid acquires different characteristics such as higher bioavailability and greater stability than Rv [36]. Similar to Rv, the glycosylated compound has antioxidant properties [37,38]. In terms of analogs formed by oligomerization, the natural dimer ε-viniferin has attracted attention for its action to reduce hyperglycemia, help the proper functioning of vascular endothelial cells and prevent atherosclerosis [39,40,41].

## 4. Methods of Extraction, Isolation and Analysis of Stilbenes

### 4.1. Extraction

Extraction of Rv from various natural sources is the first necessary step for its future use in various research directions.

A first technique involves the use of organic solvents such as acetone, ethanol, methanol, ethyl acetate or isopropyl alcohol. By investigating the optimal conditions for carrying out the technique, it was concluded that the temperature, time and type of solvent applied are the main factors that influence the recovery yield of the compound of interest. For example, in a study by Romero-Perez et al., the most efficient extraction was at 60 °C for 30 min with ethanol/water (80:20 *v*/*v*) [42]. In another experiment, the highest yield of *trans*-Rv (4.25 mg/g dw) and *trans*-ε-viniferin (2.03 mg/g) was obtained when using ethanol concentrations between 50 and 70% and a temperature of 83.6 °C [43]. Furthermore, 89.46% of Rv was extracted from peanut roots by applying deep eutectic solvent extraction with 40% of water in Choline chloride/1,4-butanediol (1/3, g/g), solid/liquid ratio 1:30 g/mL, temperature 55 °C, and extraction time 40 min [44]. Isopropyl alcohol has been used as a solvent as it is less flammable and toxic compared to other solvents. The highest amount of Rv (1.83 mg/g) and *cis*-ε-viniferin (0.65 mg/g) was obtained in the dark at room temperature [45]. A mixture of methanol and ethyl acetate [50:50 (*v*/*v*)] for 24 h at 25 °C in the dark was also used successfully in the extraction of Rv [46]. A major disadvantage of using liquid–liquid and solid–liquid extraction is that the solvents used are difficult to remove from the extract obtained and can have negative effects on human health. In this case it is necessary to perform additional purification steps that are time consuming and costly [47].

Enzymatic hydrolysis is another method used to extract stilbenes. For example, treatment with a mixture of exo-1,3-β-glucanase and pectinases at 50 °C for 60 min on the grape peel increased the amount of Rv extracted by 50% compared to extraction with ethanol-water mixture (80:20, *v*/*v*) [48]. Ionic liquid and cellulose-pectinase cocktail were successfully combined to extract Rv from tree peony seed oil extracted residues in situ [49]. In another experiment, the higher amount of Rv O-glycosides in the wine was extracted under the action of the enzyme β-glucosidase, at a temperature of 50 °C and for a period of 9 h [50]. Although there are some considerable advantages of the method (reduced numbers of stages, short execution time, temperatures not exceeding 50 °C), enzyme-assisted extraction is expensive and is not very popular in the industrial approach [51].

Supercritical carbon dioxide (SC-CO_2_) is a popular method nowadays because it is “green”: low cost, non-toxic, has a low critical point and a high diffusivity. The only impediment is that CO_2_ is a non-polar molecule, so a cosolvent such as methanol, ethanol or water is needed to allow the extraction of the compound of interest [52]. Rv was extracted by this method from *Polygonum cuspidatum* by applying the following parameters: pressure of 25.34 MPa, temperature of 51.8 °C, and ethanol content of 110.83 mL/L. Finally, an amount of about 2.5 mg/g of compound was obtained [53]. The same plant was subjected to the extraction process by using 5% acetonitrile as a solvent, at a pressure of 40 MPa and a temperature of 100 °C. After an optimal time of 45 min, resveratrol and piceid were identified [54]. The ethanol as cosolvent, but this time in a concentration of 5% (*v*/*v*), was used to extract Rv from stems, seeds, skin and pomace of the *Palomino fino* grape variety at a pressure of 400 bar and a temperature of 35 °C [55].

Another technique currently used for the extraction of natural compounds is ultrasonic or microwave assisted extraction. Ultrasonication-assisted extraction was used to extract *trans*-resveratrol from the pulp, skin and industrial waste of red grapes. Optimal conditions for the experiment were a mixture of solvents (ethanol: PEG: water ratio of 48:32:20), a temperature of 60 °C and 19.4 min for sonication time. Interestingly, the presence of PEG increased the extraction efficiency by 39.48% [56]. The microwave-assisted extraction method has been shown to be fast, with a repeatability and reproducibility >90%. Stilbene extraction from grape canes was performed under the following optimal conditions: 750 W for microwave power, 125 °C for extraction temperature, 80% ethanol in water for solvent extraction, 5 min for extraction time and sample-solvent ratios between 1:100 and 1:125 [57].

### 4.2. Isolation

The next step in Rv processing is the separation from the rest of the compounds in the extract. The literature provides many examples of Rv isolation approaches.

Wang et al. identified *trans*-resveratrol and two glucuronides by reverse-phase high-performance liquid chromatography (HPLC). A C18 preparative column was used and elution was done with 0–50% acetonitrile containing 0.1% trifluoroacetic acid, within 25 min, at 40 °C. At a flow rate of 1 mL/min, the detection was performed at 320 nm [58]. Cation-exchange chromatography was used to separate the stilbenes from 120 samples of white and red wines based on a column of resin and 75% methanol as eluent [59]. The purification of Rv was also done by means of glass columns packed with mesoporous carbon which were subsequently washed with deionized water and eluted with aqueous ethanol [60]. In another experiment, a solvent system consisting of heptane/ethyl acetate/ethanol/water (4:5:3:3, *v*/*v*/*v*/*v*) was chosen to separate the stilbenes from a compound of hairy root cultures of Peanut (*Arachis hypogaea*) by centrifugal partition chromatography [61]. A rapid method of isolating Rv from peanut press waste is molecular imprinting solid phase extraction. Polypropylene columns and reagents such as methanol, ethanol, acetone and acetic acid are used to separate the target compound [62]. Wang et al. compared two Rv purification techniques. The first, silica gel column chromatography, led to a purity of 73.70% and a recovery rate of the compound of 61.60%. The second method, called macro-porous adsorption resin-mixed bed, gave superior results: 87.12% purity and 81.10% recovery rate [63].

### 4.3. Analysis

Qualitative and quantitative analysis of compounds in a complex can be done by chromatography technique. Moreover, this technique is often used in order to extract, separate and purify the target compound. To this end, numerous types of chromatography have been developed. The principle of the method is to use a system consisting of a mobile phase and a solid phase that contribute to the separation of molecules according to size, chemical charge, shape and affinity for the stationary phase [64].

HPLC and liquid chromatography (LC) can be coupled with tandem mass spectrometry (MS/MS) for better analysis of target compounds. A mass spectrometer is needed to elucidate the identity of an unknown compound. The spectrometer can be seen as a “smasher molecule” that measures molecular and atomic masses of whole molecules, molecular fragments and atoms by generation and detection of the corresponding gas phase ions, separated according to their mass-to-charge ratio (m/z) [65]. Resveratrol has been detected and characterized by HPLC-MS, HPLC-MS/MS, LC-MS and LC-MS/MS in numerous studies [66,67,68,69,70]. Detection of stilbenes from a complex of compounds was also done by a new and fast technique called ultra-performance liquid chromatography-quadrupole time-of-flight mass spectrometry (UPLC-QTOF-MS) [71,72].

The fluorescence and phosphorescence of stilbenes are properties that can be exploited for their analysis. Two different techniques have been validated to analyze the presence of Rv in dietary supplements. The results helped the researchers conclude that constant wavelength synchronous spectrofluorimetry is faster, more sensitive and cheaper, being recommended for the precise identification of a single compound in a complex, while HPLC with fluorescence and variable wavelength detectors is characterized by sensitivity and better linearity, being recommended for determining more stilbene in a complex [73]. Similarly, high-pressure liquid chromatography with diode array detection (HPLC–DAD) and high-pressure liquid chromatography with UV detection (HPLC–UV) have been used successfully [74,75,76,77,78]

Gas chromatography (GC), another method of analysis, consists in separating compounds from a mix, based on volatility, as they are moved through a long column by a carrier gas, usually helium or nitrogen [79]. Particularly for stilbene, this method may be coupled with MS for its analytical use [80,81,82]. Another technique that can discriminate between the *cis* and *trans* isomers of Rv is capillary electrophoresis (CE). Advantages such as high efficiency, low test volume and speed support the separation of compounds from a mix based on differentiated migration—resulting from intrinsic differences in mass to charge ratios—in the electric field [83]. A summary of the main techniques for analyzing stilbenes from different natural sources is provided in Table 1.

## 5. The Stability of Resveratrol

Natural compounds have been effective in teaching us about chemical functionality that is compatible with the aqueous milieu of biological microenvironments [97]. In this matter, resveratrol is a widely studied stilbene. Over the last decades, properties such as anti-cancer, anti-inflammatory, anti-diabetic, anti-obesity, neuroprotective, antioxidant, anti-aging, cardioprotective and antiviral have been demonstrated in numerous experiments [98]. However, before evaluating the therapeutic potential of Rv, it was necessary to conduct studies on the safety, stability and bioavailability of the molecule.

The *trans* isomer of Rv is more stable and biologically active. It can be transformed into the *cis* isomer under the influence of light, pH and temperature [25]. One reason why the *cis* conformation is more unstable is the presence of the two center chain hydrogen atoms located on the same side [26]. Another reason is that in basic conditions the deprotonation of the molecule occurs, followed by a process of self-oxidation and degradation or polymerization. Therefore, the *cis* form is stable only at neutral pH and in the absence of light [99]. On the other hand, the *trans* isomer is stable at acidic pH, room or body temperature and limited exposure to oxygen or light [100]. Following the thermal characterization of *trans*-resveratrol, a stability of approximately 30 min at 70 °C has been demonstrated, while at higher temperatures the degradation of the molecule occurs very rapidly [101]. The results of a study showed that exposure of stilbenes to sunlight and UV light has triggered, over time, the process of dimerization and photoisomerization [45]. Several characteristics of Rv were tested in an experiment by Robinson et al. First, resveratrol demonstrated the best solubility in alcohol (87.98 mg/mL) and PEG-400 (373.85 mg/mL) and the weakest in water (0.05 mg/mL). Furthermore, it has been shown that the polyphenol is stable under acidic to neutral conditions for up to 193 h. Finally, a strong binding of the compound to human plasma proteins at 37 °C (98%) was calculated [102].

## 6. The Bioavailability and Safety of Resveratrol

The bioavailability and absorption of this polyphenol have been intensely studied over the last two decades. After conducting numerous in vitro and in vivo experiments in animal models and subsequently in human subjects, the researchers concluded that the bioavailability of Rv is low. Although the compound has been shown to be highly effective in various in vitro studies, the results should be interpreted with caution when switching to in vivo studies. The main explanation for this is that once in the body, resveratrol is rapidly metabolized in the liver and intestine to glucuronide and sulphate derivatives [36]. Moreover, due to the metabolic activity of the intestinal microflora and the liver functions that differ from person to person, there is great variability in the processing of the stilbene [103]. Briefly, due to the acidic environment, Rv remains stable in the stomach. Its biotransformation takes place at intestinal and hepatic levels through glucuronidation or sulfation. Under the action of sulfotransferases (SULTs) resident in the intestine and liver, the following major compounds are formed: resveratrol-3-O-sulfate, resveratrol-4′-O-sulfate and resveratrol-3, 4′-O-disulfate. The UDP-glucuronosyltransferase (UGT) family of enzymes catalyzes the glucuronidation of resveratrol to resveratrol-3-O-glucuronide and resveratrol-4′-O-glucuronide. Furthermore, the gut-resident bacteria can transform Rv into piceid (resveratrol-3-O-beta-glucoside), an important monoderivative that can be absorbed into the bloodstream. Piceatannol, 3,4′-dihydroxy-*trans*-stilbene, dihydroresveratrol (DHR) and 3,4′-dihydroxybibenzyl (lunularin) are other derivatives produced under the action of the microbiome. The obtained derivatives are bound to plasma proteins to be transported through the bloodstream. The most frequently used carrier molecules are hemoglobin and albumin, the latter demonstrating a higher binding affinity [104]. A schematic overview of the process of Rv metabolism in the human body can be found in Figure 3.

Finally, most of the compounds are eliminated from the body through urine or feces, proving that Rv has a poor bioavailability. According to some studies, the aglycone form of Rv recorded an insignificant recovery rate, while the sulfonated or glucuronidated compounds were recovered, in feces and urine, in a proportion of 71–98% after oral administration and, respectively, 54–91% after intravenous administration [105]. On the other hand, in another experiment, the presence of sulfation and glucuronidation metabolites of Rv was demonstrated at the intestinal level and the role of resveratrol-3-O-sulfate in maintaining the gut barrier function was highlighted [106].

The pharmacokinetics of Rv in humans have been studied in vivo in healthy individuals. The first experiments, performed in the early 2000s, aimed at the oral administration of a single dose of 25 mg of Rv or a dose of 0.2 mg administered intravenously and the analysis of the metabolic processing of the compound. In both cases, very small amounts of polyphenol were detected in plasma approximately 30 min to 1 h after administration (491 ng/mL and 416–471 μg/L, respectively), most being found in the urine as glucuronides and sulfate conjugates [107,108]. In another experiment, peak plasma levels of Rv (539 ± 384 ng/mL) after a single dose (0.5, 1, 2.5 or 5 g) were detected after 1.5 h, while urinary excretion of the compound and its metabolites was 77% in the first 4 h after consumption [109]. Interestingly, there are studies that suggest that a meal may decrease the bioavailability of Rv. The hypothesis was tested by consuming a single dose of 400 mg stilbene and subsequently measuring the plasma concentration of the compound [110] or 2000 mg twice daily with a high-fat breakfast [111]. In contrast, the consumption of a moderate amount of wine (300–600 mL) associated with a diet rich in nutrients showed that the bioavailability of Rv was not affected by the type of food consumed or the quantity or quality of food lipids [105]. Additionally, the administration of resveratrol in multiple doses has been identified as safe for healthy human subjects. Specifically, an amount of 25, 50, 100 or 150 mg, six times/day, for thirteen doses showed a half-life of 2–5 h and better bioavailability in the morning [112]. Therefore, considering the set of in vivo experiments performed on humans, there are some generally accepted conclusions. First of all, taking a single dose of Rv cannot have beneficial effects as its bioavailability is very low. On the other hand, administration in multiple doses may result in the accumulation of Rv in tissues and organs and the activation of its therapeutic potential, as evidenced by good plasma concentrations of the compound. Secondly, it remains to be determined whether consuming Rv with meals helps its absorption or is an impediment. Third, there is a debate about the potential adverse effects of long-term use of Rv. Adverse effects such as nausea, abdominal pain, diarrhea at 2.5 g and 5 g ingested for 29 days by healthy volunteers have been reported [113]. In another study performed on schizophrenic and obese patients, Rv treatment worsened their lipid profile and raised the cholesterol levels [114]. Further studies are needed in this regard because definitive conclusions on Rv toxicity cannot be drawn yet.

In conclusion, given the existence of contradictory and inconclusive data in the literature regarding the safety and bioavailability of Rv, more experiments are needed in the future. However, its safety in small doses has so far been well documented in scientific articles. Moreover, the therapeutic effects of resveratrol have not been fully explored and understood so far. Parameters such as contexts of treatment, dosages and durations administered must be further tested in order to develop a correct overview of the polyphenol’s beneficial effects.

## 7. The Anti-Angiogenic Modulatory Effect of Resveratrol

Vascular endothelial growth factor (VEGF) is a dimeric glycoprotein with a molecular weight of 40 kDa. In mammals there is a VEGF family of seven proteins: VEGF-A (the most common), VEGF-B, VEGF-C, VEGF-D, VEGF-E, VEGF-F and PlGF (placental growth factor) [115]. VEGF-A was first isolated in 1989 and received the role of endothelial cell-specific mitogen [116]. Over time, this essential growth factor for vascular endothelial cells has been identified in other cell types, such as keratinocytes [117], hypertrophic chondrocytes [118], macrophages [119], tumor cells [120] and platelets [121]. VEGF has important roles in the development of the embryonic vascular system, vascular permeability and endothelial cell proliferation and migration [122]. In addition to its beneficial effects on the proper functioning of the body, VEGF has also been associated with the onset of the angiogenesis process. Today, the most significant biomedical implications of VEGF concern its role as a mediator of pathological angiogenesis, such as that associated with cancer and intraocular neovascular disorders [116].

An enormous burden on the healthcare system is caused by eye diseases that affect people’s quality of life. Among them, some of the most common are age-related macular degeneration (AMD) and diabetic retinopathy (DR) [123]. Moreover, given that the world’s population is aging, an increase in patients suffering from one of these eye diseases is expected. A meta-analysis conducted in 2019 estimated an increase in people with diabetes affected by any diabetic eye disease in Europe from 6.4 million to 8.6 million in 2050. It was also predicted the presence of any diabetic eye disease in more than a quarter of persons with type 2 diabetes and half of persons with type 1 diabetes [124].

The first clinical hallmark of AMD is the presence of drusen (drusen bodies)—small yellow deposits of lipids, proteins and cellular debris that accumulate under the retina and cause damage to the retinal pigment epithelium (RPE) cells. Subsequently, a chronic aberrant local inflammatory process begins that leads to atrophy of certain areas of the retina and the expression of inflammatory cytokines such as VEGF. These processes are accompanied by the onset of choroidal neovascularization and increased vascular permeability and fragility [125]. AMD can be divided into two categories depending on its evolution: the first stage is dry AMD (accounts for ∼90% of cases), and the second is wet AMD—the late stage which is characterized by excessive production of VEGF [126]. DR, a disease strongly associated with diabetes and the condition of hyperglycemia, is manifested by the following clinical signs: vascular leakage, pericyte loss, weakening of the inner blood-retinal barrier and compromised capillary integrity. From a biochemical point of view, the main manifestations are generation of oxidative stress, activation of protein kinase C pathway, increased generation of advanced glycation end products (AGEs) and hyperglycemic pseudohypoxia [127]. DR can be non-proliferative or proliferative, the latter being characterized by neovascularization originating from the retina and/or optic disc in patients with diabetes mellitus [128]. Non-proliferative DR is usually asymptomatic and has several well-defined characteristics: intraretinal microvascular abnormalities, microaneurysms, intraretinal hemorrhages and the potential to become proliferative over time [129] (Figure 4).

Currently, DR management is done through several approaches: laser photocoagulation, vitrectomy and intravitreal injection with anti-VEGF agents or corticosteroids (dexamethasone, triamcinolone acetonide). Ranibizumab (Lucentis^®^, Novartis, Basel, Switzerland), the off-label bevacizumab (Avastin, Genentech, CA, USA), and aflibercept (Eylea, Regeneron, NY, USA) are the anti-VEGF agents from the “first-line treatment”. Aflibercept and ranibizumab have been approved by the United States Food and Drug Administration (FDA) [131,132,133]. Bevacizumab is a humanized monoclonal antibody that specifically binds to the VEGF-A protein [134]. Recently, its use by intravitreal administration in patients with central retinal vein occlusions (CRVO) improved best-corrected visual acuity (BCVA) and reduced the degree of cystoid macular edema (CME) after a 5-year treatment period [135]. Aflibercept is a fusion protein with high VEGF affinity attributed to binding sequences from the native receptors VEGFR1 and VEGFR2 [136]. Ranibizumab is a recombinant, humanized, monoclonal antibody fragment (Fab) with a molecular weight of 48 kDa that inhibits the action of VEGF-A and is produced by an *E. coli* expression system in a nutrient medium containing the antibiotic tetracycline [137]. However, these therapeutic agents also have side effects that must be taken into consideration when using them. The most common side effects were summarized by Cheung et al. in their work: hypertension, proteinuria, and impaired wound healing, repeated administration because of the agent’s short half-life, vitreous hemorrhage, infection, potential loss of neural retinal cells, retinal detachment, cataract formation and the passage of the anti-VEGF agent into the systemic circulation [138]. The treatment of retinopathy of prematurity, characterized by incomplete or non-normal retinal vascularization, with ranibizumab succeeded in suppressing the VEGF level only temporarily, with the risk of recurrence and the formation of a retinal peripheral avascular area [139].

Attempts are currently being made to develop therapeutic strategies that will overcome the obstacles encountered by current treatments. One of the research directions implemented by scientists is the use of Rv as a potential therapeutic agent in the treatment of DR. It is well known that this polyphenol can pass the blood–brain barrier (BBB) via transmembrane diffusion [140,141], this being essential in its use as a new therapeutic agent in the treatment of eye diseases. Specifically, the anti-angiogenic action of Rv has been demonstrated in numerous in vitro and in vivo studies.

In an experiment in which human retinal pigment epithelial cells (ARPE-19) were subjected to a glucose concentration of 33 mM (in order to simulate the condition of hyperglycemia which is characteristic for diabetes) and subsequently treated with Rv in different concentrations (1.25 μM, 2.5 μM, 5 μM and 10 μM), the amount of VEGF expressed decreased significantly in a dose-dependent manner [142]. In another study, an increase in VEGF expression was made artificially by exposing ARPE-19 cells to cobalt chloride, a hypoxia mimetic agent (100 μM). Thereafter, they were treated with Rv (10–50 μM) and the level of VEGF protein was assessed by ELISA. According to the data obtained, a significant inhibition of VEGF production was recorded at Rv concentrations of 10 μM, 20 μM and 50 μM [143]. Exposure of human retinal pigment epithelial cells for 24 h to Rv (2–50 μM) has been shown to have a significant anti-angiogenic effect by reducing the amount of VEGF-A and VEGF-C produced [144].

In an in vivo approach, seven-day-old Sprague-Dawley rats with oxygen-induced retinopathy of prematurity were divided into several groups and treated with Rv (10, 30, and 60 mg/kg/day). Western blot analysis showed a decrease in VEGF expression in a dose-dependent manner of Rv: 3.4%, 23.0%, and 43.7%, respectively [145]. The corresponding VEGF microRNA level was investigated in twenty-four adults male Wistar albino diabetic rats who received an intraperitoneal Rv injection (10 mg/kg/day) for 4 consecutive weeks. The results of the study showed that Rv treatment did not significantly influence the level of expression [146]. In another study, diabetes induced C57BL/6 mice were treated for 4 weeks with 20 mg/kg of Rv administered orally. VEGF levels were inhibited after the treatment, according to WB analysis [147]. The decrease in VEGF production was revealed by ELISA in Sprague-Dawley (SD) male rats (14 weeks old) treated with an intravitreal injection of Rv (0.1 µg/mL or 1 µg/mL) in one eye while the other one served as a vehicle [148].

## 8. The Antioxidant Role of Resveratrol in Eye Diseases

An imbalance between production and accumulation of oxygen reactive species (ROS) triggers oxidative stress (OS). Even though cells benefit from non-enzymatic (glutathione, tocopherols) and enzymatic antioxidant systems such as glutathione peroxidase (GPx), superoxide dismutase (SOD), and catalase (CAT), large amounts of ROS can cause cell and tissue damage [149]. The retina is susceptible to ROS because of high-energy demands and exposure to light. Thus, when the ROS balance is broken, retinal cell injury appears because ROS begin to interact with the cellular components [150].

Various in vitro studies have been performed to verify the antioxidant capacity of Rv in the treatment of AMD and DR. For example, ARPE-19 cells grown under H_2_O_2_-induced intracellular oxidation conditions and subsequently treated with Rv (100 μmol/L) demonstrated a decrease in proliferation and oxidative stress-induced cell death [151]. The action of Rv to activate the AMPK/Sirt1/PGC-1α pathway that causes a decrease in intracellular ROS concentration was tested on bovine retinal capillary endothelial cells (BRECs) grown under simulated DR conditions (30 mM glucose) and then treated with Rv in concentration of 20 μM [152]. Human retinal D407 RPE pretreated with Rv at different concentrations (25, 50 and 100 μM) for 24 h and exposed for 1 h to 500 μM hydrogen peroxide were analyzed for the antioxidant capacity of the stilbene. The results indicated enhancing antioxidant enzyme activities for SOD, GPx and catalase and a reduction in the generation of intracellular ROS [153]. The antioxidant effect of Rv was reviewed by Lancon et al. (2016), suggesting that this natural antioxidant molecule increases the activity of the defense enzymes, thus limiting the formation of ROS in various cell types of the eye, protecting them against an irreversible damage [154]. In an example performed on porcine primary cells belonging to the trabecular meshwork (TM) tissue grown in oxidative stress (40% O_2_) and treated with a concentration of 25 μM Rv, a decrease in the characteristic markers for primary open-angle glaucoma (POAG) was obtained: IL1α, IL6, IL8, ROS, senescence markers sa-β-gal and lipofuscin [155]. Similarly, Rv (different applied concentrations ranging from 2.5–20 µM) protected human lens epithelial cell line (HLEB-3) against H_2_O_2_ induced oxidative damage by inducing an increase in the concentration of the antioxidant enzymes SOD, catalase and HO-1 and by inhibiting the activation of p38 and JNK, molecules involved in cell death pathway [156]. A summary of the discussed in vitro experiments that highlights aspects of the experimental design could be found in Table 2.

Although there are not many in vivo studies, interesting results have been obtained in several experiments. Oral administration of Rv for four months (5 mg/kg/day) in male Wistar diabetic rats improved blood SOD activity (*p* < 0.05) and significantly decreased NF-kβ activity and apoptosis rate [157]. In another identical experimental study, the amount of plasma oxidative stress markers decreased significantly (nitrite/nitrate, GSSG/GSH ratio) after comparing diabetic mice with those treated with Rv [158]. *Trans*-resveratrol (5 mg/kg body weight) alleviated oxidative stress as indicated by reduced 4-hydroxynonenal (4-HNE) in Dark Agouti diabetic rats and by the recovery in lipid peroxidation (*p* < 0.05). In comparison, the oxidative stress marker increased in diabetic rats (*p* < 0.05) [159]. VEGF-activated endothelial NO synthase (eNOS) is a nitric oxide (NO) generating enzyme involved in retinal and choroidal neovascularization. In an in vivo experimental model, Rv treatment with 10 mg/kg/day administered to Wistar albino diabetic rats decreased eNOS mRNA expression and nitrite—nitrate levels in eye tissue compared to diabetic control animals [146].

**Table 2 antioxidants-11-02447-t002:** Experiments performed on retinal cell lines aiming to investigate the potential therapeutic effect of resveratrol in different doses administered.

Resveratrol Concentration	Cell Line	Method/Target Molecule	Effect	Reference
10 mM—inside microcapsules	D407 RPE	ELISA	↓ VEGF ↓ IL-6	[160]
25, 50, and 100 μM	D407 RPE	The activity of antioxidant enzymes	↑ SOD ↑ Catalase ↑ Reduced glutathione	[153]
12.5, 25, 50 and 100 mg/L	D407 RPE	ELISA Western Blotting qRT-PCR	↑ SOD ↓ MDA ↑ Bcl-2 ↓ Caspase-3	[161]
10 μM	ARPE-19	SIRT1 Activity Assay Kit DNMT Activity Quantification Kit qRT-PCR	↑ SIRT1 levels ↑ DNA methyltransferases (DNMTs)	[162]
10 μM	ARPE-19	Flow cytometry Acridine orange staining	↓ Cell death ↑ Autophagy	[163]
10 μM	ARPE-19	ELISA	↓ VEGF	[164]
10 μM	ARPE-19	ELISA	↓ VEGF ↓ IL-6 ↓ IL-8	[142]
10 μm 20 μM 50 μM	ARPE-19	ELISA Western blotting	↓ VEGF	[143]
50 μM 100 μM	ARPE-19	Western blotting	↓ ERK 1/2	[151]
2–50 μM	ARPE-19	ELISA RT-PCR	↓ VEGF-A ↓ VEGF-C	[144]
40 μM	E1A.NR3 retinal cells	Western blotting	↑ SIRT-1 ↑ Ku70 ↓ Bax	[165]
Pre-treatment for 24 h with 1 μM	HRECs	Carboxy-DCFDA	↓ Intracellular ROS levels	[166]
5 μM	HUVECs	Western blotting PCR	↑ SIRT1 levels	[167]
25 μM	Primary porcine trabecular meshwork cells	Carboxy-DCFDA RT-PCR Flow cytometry	↓ Intracellular ROS levels ↓ IL1α, IL6, IL8, ↓ sa-β-gal and lipofuscin	[155]
2.5 μM, 5 μM, 10 μM and 20 μM	Human lens epithelial cells (HLEB-3)	WST-1 Flow cytometry Western blotting	↑ Cell viability after H_2_O_2_ damage ↓ p38 and JNK phosphorylation ↑ SOD, Catalase and HO-1 expression	[156]

↑—Overexpression/Increased level; ↓—Down-regulation/Decreased level.

## 9. The Relationship between Resveratrol and Sirtuins

Sirtuins are an ancient class of proteins present in organisms from yeasts and bacteria to eukaryotes that play an important role in metabolic regulation. They are part of class III histone deacetylases (HDAC) and need NAD+ as a co-substrate in the reactions in which they participate. In mammals there is a family of seven sirtuins (SIRT 1-7) of which SIRT1 (silent information regulator 2) is the most studied currently. The catalytic center of the protein contains 275 amino acids and has an affinity for histones, some plasma proteins and transcription factors [168]. The roles that sirtuins play in the human body are very diverse, such as fetal development, regulation of energy metabolism, oxidative stress response, chromatin structure and transcription apoptosis, and cell survival [169,170]. These deacetylases have been linked to the appearance of pathologies such as neurodegenerative diseases and various types of cancer (melanoma, glioma, breast, colorectal, liver, leukemia, etc.) [171]. Consistent with the functions they perform, their location is different: SIRT1—nucleus and cytosol, SIRT2—cytosol, SIRT3, SIRT4 and SIRT5—mitochondria, SIRT6—nucleus and SIRT7—nucleolus [172].

There are interesting studies that correlate the action of sirtuins with the management of DR. For example, SIRT1, 3, 5 and 6 are key enzymes in DR since they modulate glucose metabolism, insulin sensitivity and inflammation [173]. Moreover, in accordance with the cumulative evidence, SIRT1 and SIRT6 repress pancreatic β cell dysfunction, attenuating the development of type II diabetes [174,175]. Regarding the link between these proteins and Rv, it has been shown that there is a molecular model by which the polyphenol can directly activate sirtuins [176] (Figure 5).

An in vitro study was performed on human retinal vascular endothelial cell (HRVEC) exposed to 30 mM glucose and treated with Rv at a concentration of 10 μmol/L. Subsequently, the inflammatory process triggered by exposure to high glucose concentrations was attenuated by Rv through the activation of the SIRT1/AMPK/NF-κB signaling pathway [177]. E1A.NR3 retinal cells pretreated with 40 μM Rv recorded a 5-fold increase in the amount of SIRT1 expressed compared to cells exposed to anti-recoverin (Rec-1) and anti-enolase (Enol-1), antibodies linked to cancer-associated retinopathy [165].

An in vivo study focused on the activity of sirtuins in the following experimental design: rats with type I diabetes treated with Rv (30 mg/kg/day) and rats with type II diabetes treated with Rv in different concentrations (10 mg/kg/day), for 8 weeks. There was a significant increase in the level of microRNAs corresponding to SIRT-1, 2, 3 and SIRT-5 for type I diabetes and for SIRT-1, 2 for type II diabetes, respectively. On the other hand, there was a decrease in the microRNA level of SIRT-3 in type II diabetes [178]. Rv has previously been shown to be an activator of adenosine monophosphate (AMP)-activated protein kinase (AMPK). Systemic administration of Rv (50 mg/kg for 7 days) in STZ-induced diabetic mice models triggered the activation of AMPK and thus the suppression of NF-kB, a transcription factor involved in the regulation of the genes of some pro-inflammatory molecules. SIRT1 activity also increased significantly after treatment [179]. Retinal SIRT1 activity, reduced in BALB/c male mice with light damage, was significantly augmented by Rv treatment at a dose of 50 mg/kg body weight for 5 days [180].

There is a large body of evidence to suggest that Rv has various mechanisms of action that can help treat DR. Whether it acts on pro-inflammatory markers, on enzymes involved in oxidative stress or on sirtuins, this polyphenol is of interest to the scientific world. However, in order to make the most of its therapeutic potential, it is necessary to overcome major disadvantages such as low bioavailability, solubility and stability. A good solution is to encapsulate Rv in various micro or nano transport systems that can be delivered targeted. Therefore, in the next chapter we will discuss key aspects of these systems as well as their applications in encapsulating Rv.

## 10. Resveratrol Delivery Systems

Even if resveratrol exhibits a broad spectrum of promising therapeutic activities, it remains unclear whether after its administration in pure form it can reach the target site. The short biological half-life of the compound and its transformation in the digestive system made its in vivo applicability a limited success [181]. To overcome the main disadvantages identified in the biological use of resveratrol researchers have implemented the use of delivery systems (Figure 6).

Encapsulation is a technique in which one or more ingredients are trapped within some form of matrix—microscopic or macroscopic, solid or liquid, homogenous or heterogenous. The encapsulated compound is considered the “active” part and the material from which the coating is made is called “wall” or “carrier”. Usually, a delivery system can have nanometric (1–1000 nm) or micrometric (1–1000 μm) dimensions [182]. There are three main reasons why Rv is encapsulated. First, this technique is needed to enhance the oral bioavailability by increasing its solubility in gastrointestinal fluids, promoting its absorption by enterocyte cells, and reducing its metabolism prior to absorption. Second, the barrier to low solubility of Rv in aqueous solutions must be overcome. Third, Rv needs to be protected from environmental factors that promote its chemical degradation such as ultraviolet light, pH, temperature and oxygen [183].

The main nanocarrier systems used in Rv encapsulation are solid lipid nanoparticles (SLNs), liposomes, niosomes, dendrimers, micelles and nanocapsules (Figure 6).

SLNs are lipid-based delivery systems that occur in many sizes in the range from 30 to 1000 nm [184]. Their structure includes a mix of solid lipids stabilized by surfactants in an aqueous media. SLNs and nanostructured lipid carriers (NLCs) loaded with Rv were successfully produced to improve the bioavailability of the compound. With a size between 150 and 250 nm and an almost uniform spherical shape, the nanoparticles demonstrated an Rv encapsulation efficiency of 70% [185]. Rv loaded SLNs (248 nm) were dissolved in distilled water and administered orally by gavage daily in rats with type 2 diabetes for 1 month. The results of the study showed an improvement in insulin resistance through the upregulation of SNARE protein complex, suggesting that this type of transport of the polyphenol may be a therapeutic method of interest [186].

Another nanosystem that has gained attention in the scientific world consists of an aqueous core surrounded by one or more layers of phospholipids and cholesterol that form a lipid bilayer and are called liposomes [187]. With a structure similar to that of cell membranes, this nanosystem is much more biocompatible than other synthetic materials. Moreover, its surface can be easily functionalized, for example with PEG, to facilitate targeted delivery [187]. Rv loaded liposomes showed promising results regarding their therapeutic potential when tested in vivo for the management of breast cancer [188], glioblastoma [189], hepatocellular carcinoma [190] and Parkinson’s disease [191]. In an experiment related to DR, 200 nm Rv-loaded nanoliposomal formulations showed prolonged antioxidant activity against oxidative stress for 24 h and reduced the glucose levels significantly in pancreatic β TC cells [192].

Niosomes are vesicles consisting of an aqueous core enclosed within a non-ionic surfactant bilayer, thus forming a closed bilayer structure. They are similar to liposomes from a structurally point of view, being a stable and less expensive alternative version of those [193]. On the other hand, there is a possibility that the vesicles may fuse and the encapsulated drug may leak or be hydrolyzed. Moreover, the synthesis process of niosomal carriers is time consuming and requires special equipment [194]. Good ocular tolerance of Rv loaded chitoniosomes formulation without any inflammatory response to the rabbit eyes was obtained by El-Haddad et al., 2021. With a therapeutic encapsulation efficiency of 85% and a size below 500 nm, they were tested in vivo on adult male albino rabbit. The anti-inflammatory effects after 3 days of treatment revealed reduced gene expressions of TNFα and IL-6, down to 49% and 55% respectively, in treated groups compared to control group [195].

Dendrimers are synthetic polymers with a layered structure and dimensions below 15 nm. Their surface can be easily conjugated with various drugs or nucleic acids [196]. They are mainly used in the delivery of genes and drugs, having the advantage of being connected to liposomes, carbon nanotubes or various nanoparticles. Dendrimers are biocompatible and can be easily eliminated from the body. In contrast, they may exhibit cytotoxicity that contributes to the destruction of normal cells [187].

This system named micelles is made of amphipathic linear polymers formed spontaneously by self-assembly in water and a hydrophobic core in which the drug is encapsulated [197]. With a size ranging from 20 to 100 nm, they have a low volume of distribution and accumulate in body areas with compromised vasculature where they behave like carrier systems for drug targeting as they can carry specific ligands on their surface [198]. According to in vivo experiments, the level of expression of the inflammatory markers IL-6, TNFα and COX-1 decreased significantly after treatment of the inflamed cornea with micelle ophthalmic solution containing resveratrol. At the same time, the mRNAs corresponding to the antioxidant enzymes SOD and haem oxygenase 1 (HO-1) and the protein SIRT1 increased significantly under the same experimental conditions [199].

Nanocapsules is a commonly used delivery system that comes in various forms: polymeric nanocapsules, silica nanocapsules, gold or silver nanocapsules. Due to the presence of numerous silanol groups on their surface, silica nanocapsules can be easily modified in order to have different functionalities [200]. Gold and silver nanoparticles demonstrated good biocompatibility, with those below 50 nm diameter having the ability to cross the blood–brain barrier [201]. Gold nanoparticles (AuNPs) containing Rv were administered orally on streptozotocin (STZ) induced diabetic rats for a period of 3 months (once/day with 200 or 300 mg / kg quantity of AuNPs in a volume of 1.5 mL/kg purified water). After this period, these AuNPs with an average size of 10 nm showed a reduction in the level of mRNA corresponding to VEGF-1, IL-6, Tumor Necrosis Factor (TNFα), Monocyte Chemotactic Proteins-1 (MCP-1) and Intercellular Adhesion Molecules-1 (ICAM-1). Furthermore, a reduction in Nf-kB phosphorylation was observed [202]. Chitosan—pectin core—shell nanoparticles loaded with Rv have been tested for their antioxidant activity compared to free resveratrol and have shown better antioxidant potential. Additionally, the release of Rv from the nanosystem was obtained in the acidic pH of the stomach but also in the alkaline pH of the intestine for almost 30 h [203]. In another example, the topical release of Rv from the natural seed butter (SLN) of *Theobroma grandiflorum* was tested. Homogeneous nanospheres obtained with a size ranged from 150 nm to 200 nm demonstrated an Rv encapsulation efficiency of approximately 74% and an increase in the antioxidant activity in a dose-dependent manner. Additionally, the release of polyphenols was done gradually, in 24 h, reaching 80.48 ± 12.20% Rv [204]. Pure *trans*-resveratrol nanoparticles with a mean size of 170 nm with an absolute bioavailability of 25.2%, tested on rats, were fabricated by a supercritical antisolvent (SAS) process [205]. Reduction of intraocular pressure was demonstrated in a normotensive eye rabbit model following the application of Rv (1 mg/mL) chitosan nanoparticles with dimensions below 100 nm [206]. The use of poly (lactic-co-glycolic-acid) nanoparticles was proposed for the delivery of resveratrol. The delivery system was efficient in reducing the expression of VEGF in ARPE-19 cells, and, therefore, could be a useful strategy for the treatment of neovascular AMD [164]. Resveratrol-loaded mucoadhesive lecithin/chitosan nanoparticles were topically instilled into the lower conjunctival sac of the left eye of albino rabbits in order to test the lower drug retention to the eye’s anterior portion, an area of primary interest for the ocular drug delivery system. The nanoparticles had an average size of 163.3 nm and an RV encapsulation efficiency of 97.03%. By interacting with the mucus layer, these nanoparticles exerted a long-term release of the therapeutic agent [207].

Microsystems are another category of Rv delivery systems that have been studied extensively in recent years (Figure 6). Like the nanosystems described above, they fall into several categories, each with its own characteristics, advantages and disadvantages.

Drugs can be encapsulated in microparticles or microspheres with targeted delivery achieved by binding ligands to their surface. So far, much of the research conducted in this area has been focused on seeking biodegradable and biocompatible polymer materials [208]. Both synthetic and natural polymers can be used individually or in different combinations to build the system’s walls [209]. In terms of encapsulation techniques, there is a rich variety of approaches: the layer-by-layer (Lbl) technique, spray drying, freeze-drying, ionic gelation, co-precipitation and coacervation [210,211,212,213,214,215]. Lecithin-polysaccharide self-assembled microspheres with a dimension of 12 μm entrapped Rv with an efficiency of 92% and maintained an elevated bioavailability of the compound during in vitro simulated digestion [216]. Rv can be added in gastro-resistant pectin-alginate-coated microcapsules with an encapsulation efficiency of 41.72% ± 1.92% and a release efficiency of the compound about 70% of the total, within 24 h and pH 7.4 [217]. An encapsulation efficiency of 96.8% was obtained for Rv entrapped between the walls of a polyelectrolyte multilayer system that was designed to be delivered inside retina pigmented epithelial cells. Having a diameter of 3.5 μm, these microcarriers were monitored inside D407 retina pigmented epithelial cells due to the fluorescent rhodamine 6G dye [160]. Another fluorescent dye, fluorescein isothiocyanate, was introduced inside near-infrared (NIR) responsive polymeric microcapsules carrying resveratrol in tandem with gold nanobipyramids to observe the internalization process in D407 cells. These microcarriers, with a diameter of 2.5 μm, were able to release the therapeutic agent and showed no cytotoxicity towards the tested cells [218].

Hydrogels, three dimensional cross-linked networks of polymer chains, are potential materials for drug delivery owing to their similarity with extra cell matrix, softness, hydrophilicity, viscoelasticity, biodegradability and biocompatibility. They can also be modified to respond to various stimuli such as light, temperature, pH, magnetic or electric field, pressure or ionic power [219]. Moreover, their toxicity is negligible [220]. Rv was encapsulated in high molecular weight chitosan-based nanogels measuring approximately 140 nm and a round overall morphology. These nanogels were rapidly internalized in ARPE-19 cells where they showed no cytotoxicity and escaped from the endo/lysosomal acidic compartments, demonstrating their safety and applicability in ocular treatments [221].

## 11. Conclusions and Future Perspectives

Despite the fact that there is a considerable number of studies that have investigated the therapeutic potential of Rv in the treatment of eye diseases, there is a gap between the scientific results that prevents the establishment of well-defined conclusions. It is necessary to focus on studies that have an experimental design in which the administered dose, treatment period and adverse effects are well documented. Moreover, knowing the mechanism of action of Rv and its target molecules are other key aspects that must be taken into account to implement a successful treatment. In improving the stability and bioavailability of Rv there are different types of resveratrol delivery systems that have been manufactured and have succeeded.

## Figures and Tables

**Figure 1 antioxidants-11-02447-f001:**
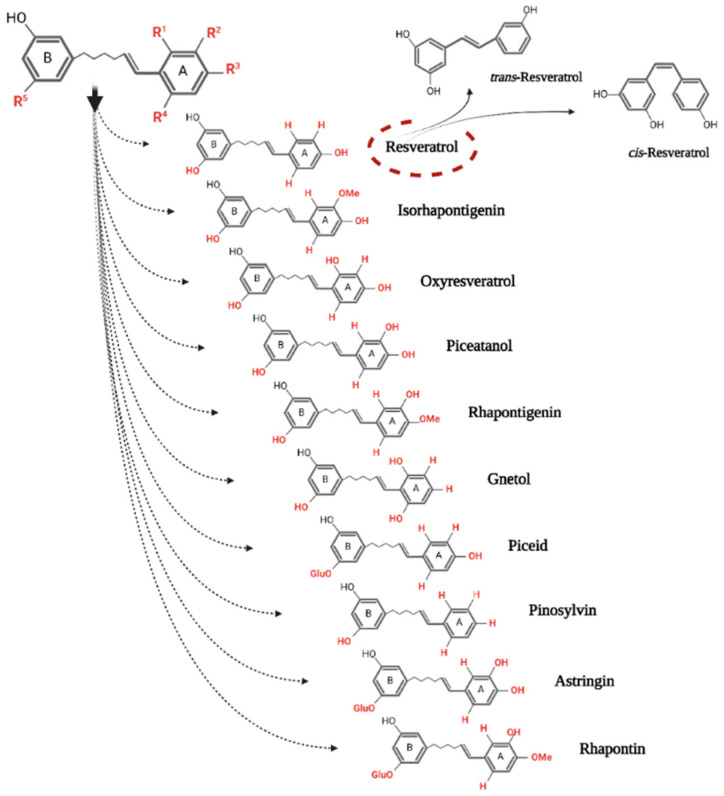
The chemical structure of the two conformations of resveratrol and its main monomeric derivatives.

**Figure 2 antioxidants-11-02447-f002:**
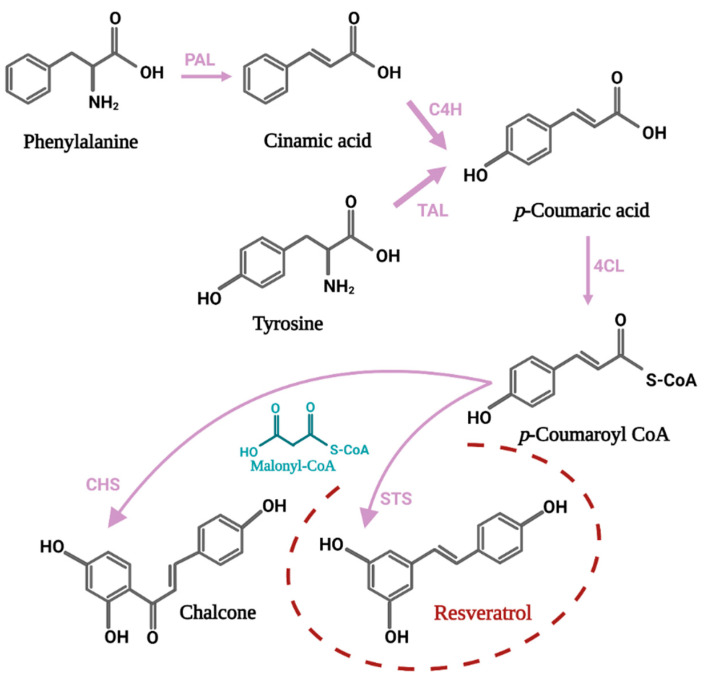
The connection between the resveratrol biosynthesis pathway and the biochemical pathway for synthesis of flavonoids, represented by the first intermediate called chalcone. The enzymes involved in the chemical reactions have been abbreviated as follows: PAL—Phenylalanine ammonia lyase; C4H—Cinnamate-4-hydroxylase; TAL—Tyrosine ammonia-lyase; 4CL—Coumaroyl-CoA ligase; STS—Stilbene synthase; CHS—Chalcone synthase.

**Figure 3 antioxidants-11-02447-f003:**
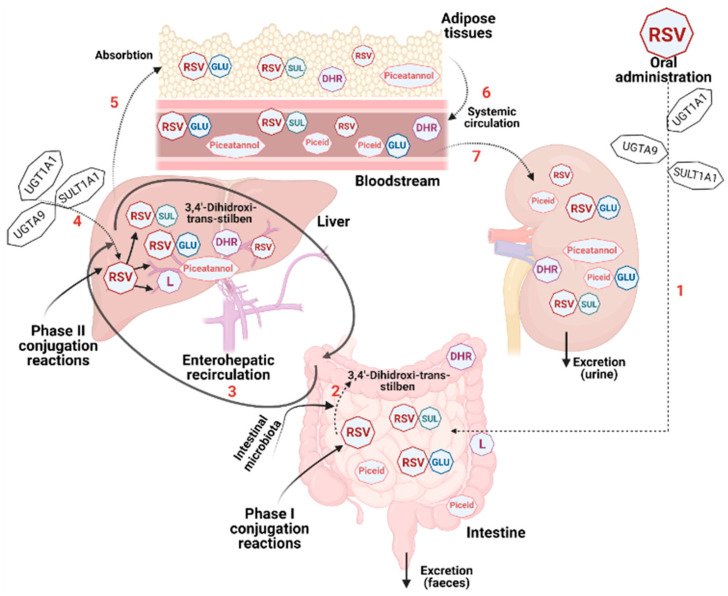
Overview of the biotransformation process of resveratrol in the human body. After oral administration, the parent molecule undergoes sulfation by SULT1A1 in the intestine and liver (RSV-SUL). Similarly, the enzymes UGT1A1 and UGTA9, expressed at intestinal and hepatic level, catalyze the formation of glucuronidated metabolites (RSV-GLU). Piceid, piceatannol, 3,4′-dihydroxy-*trans*-stilbene, dihydroresveratrol (DHR) and 3,4′-dihydroxybibenzyl (lunularin) (L) are obtained after the gut microbiome action on the ingested compound. Finally, the compounds are absorbed into the bloodstream and are subsequently eliminated from the body through urine or feces.

**Figure 4 antioxidants-11-02447-f004:**
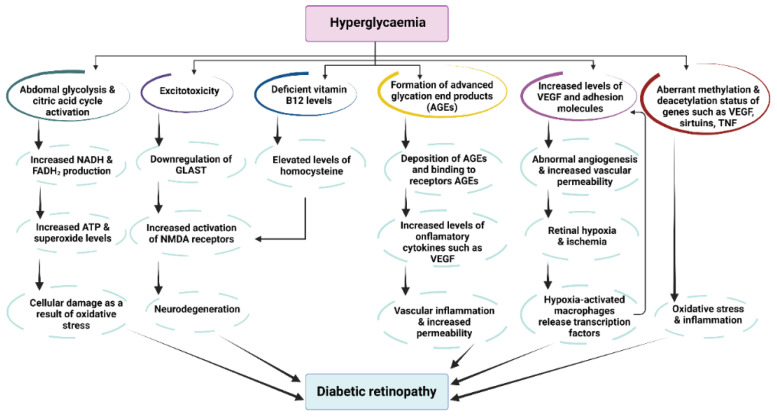
Pathophysiological pathways of diabetic retinopathy as a consequence of hyperglycemia. Abbreviations: AGEs—advance glycation end products; FADH2—redox cofactor flavin adenine dinucleotide; GLAST—glutamate transporter protein; NADH—reduced form of nicotinamide adenine dinucleotide; NMDA—N-methyl D-aspartate receptor; TNF—tumor necrosis factor; VEGF—vascular endothelium growth factor (adapted from [130]).

**Figure 5 antioxidants-11-02447-f005:**
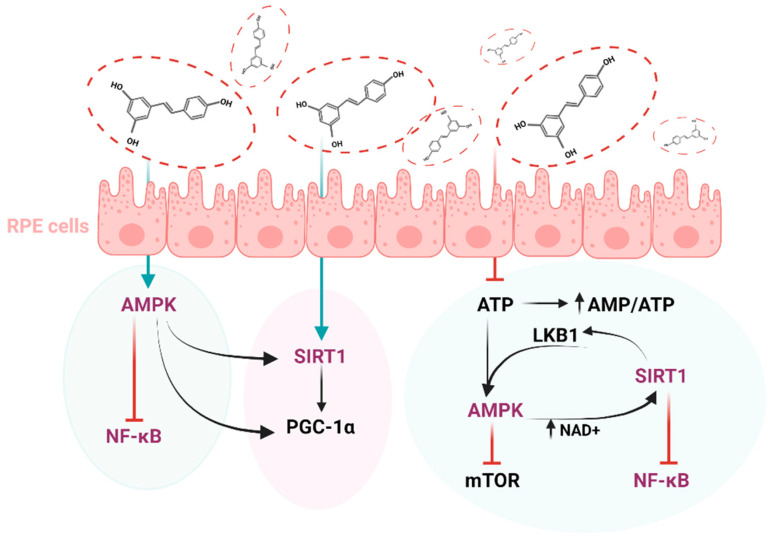
The interplay between key molecules implicated in the management of diabetic retinopathy. Abbreviations: AMPK—AMP-activated protein kinase, SIRT1—Sirtuin 1, NF-kB—Nuclear factor-kB, PGC-1α—Peroxisome proliferator-activated receptor-gamma coactivator, mTOR—Mammalian target of rapamycin, LKB1—liver kinase B1, AMP—Adenosine monophosphate, ATP—Adenosine 5′-triphosphate, NAD^+^—Nicotinamide adenine dinucleotide.

**Figure 6 antioxidants-11-02447-f006:**
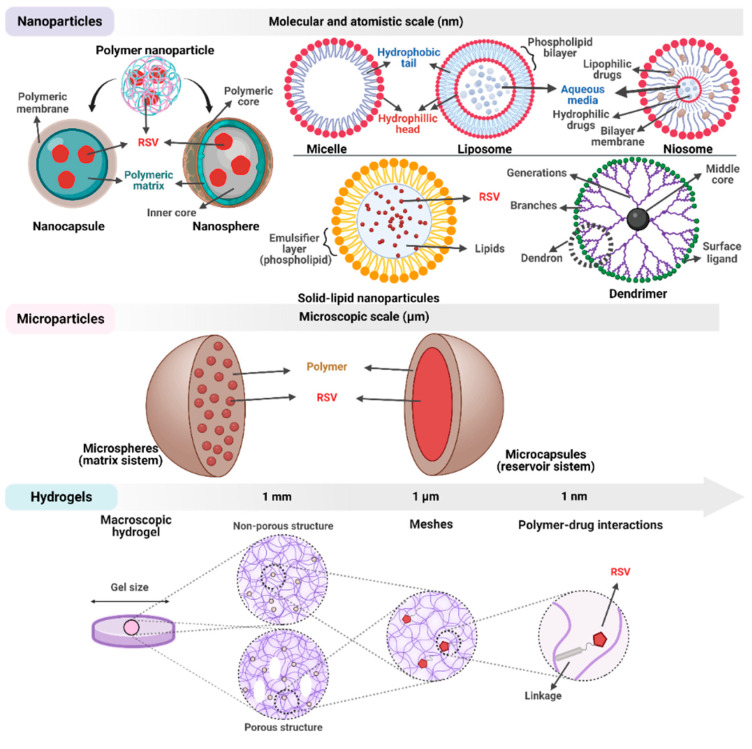
Structural aspects regarding the main resveratrol delivery systems successfully used to date.

**Table 1 antioxidants-11-02447-t001:** Methods for identifying stilbenes from natural sources used successfully in various experiments.

Analytical Method	Identified Stilbene	Source	Reference
**HPLC-MS**	*Trans*-resveratrol	Bilberries (*Vaccinium myrtillus* L.) Highbush blueberries (*Vaccinium corymbosum* L.)	[84]
	*Trans*-resveratrol	Red wines	[69]
	*Trans*-piceid *Cis*-piceid	Red seedless grape Raspberry frozen Raspberry Peach Plum Different varieties of pears and apples	[67]
	*Trans*-resveratrol	Red seedless grape Raspberry frozen Raspberry Tomato Plum	
	*Cis*-piceid *Trans*-piceid *Trans*-resveratrol ε-viniferin	Grapes (*Vitis amurensis)*	[85]
	ε-viniferin	Wild grape (*Vitis wilsonae*)	[86]
**HPLC-MS/MS**	*Trans*-resveratrol *Trans*-piceid	Dark chocolate and cocoa liquor	[87]
**UPLC-QTOF-MS**	Resveratrol Resveratrol 3-O-glucoside	Lotus (*Nelumbo nucifera*)	[71]
	*Trans*-resveratrol Vis-resveratrol *Trans*-piceid *Cis*-piceid	*Fallopia japonica* *Fallopia sachalinensis*	[72]
	*Trans*-resveratrol Resveratrol oligomers Viniferins Piceatannol	Grapes (*Vitis vinifera*)	[88]
**GC-MS**	Resveratrol Piceatannol	Berries (*Vaccinium*)	[89]
	*Trans*-resveratrol *Cis*-resveratrol *Trans*-piceid *Cis*-piceid	Tomato fruit (*Lycopersicon esculentum Mill*.)	[13]
	*Trans*-resveratrol *Cis*-resveratrol	Black tea Green tea Red tea Chamomile	[82]
	*Trans*-resveratrol *Cis*-resveratrol	Peanut (*Arachis hypogaea* L.) varieties Pistachio (*Pistacia vera* L.) varieties	[90]
	*Trans*-resveratrol *Cis*-resveratrol	Red Wines	[91]
**GC-MS/MS**	*Trans*-resveratrol *Cis*-resveratrol	Red Wines	[92]
**UPLC-MS/MS**	*Trans*-resveratrol 3-O-β-D-glucuronide 4′-O-β-D-glucuronide	Grape skins, grape seeds and grape stems	[93]
**HPLC-DAD/UV**	Resveratrol	Tissues of *Vitis amurensis*	[94]
	*Trans*-resveratrol *Cis*-resveratrol	Peanut (*Arachis hypogaea* L.) varieties Pistachio (*Pistacia vera* L.) varieties	[90]
	*Trans*-resveratrol	Nero d’Avola red grape variety	[75]
	*Trans*-resveratrol	Grapes of a wide range of *Vitis* subspecies (*sativa* and *sylvestris*)	[76]
	Resveratrol-O-sulfate Resveratrol-O-diglucoside	Rhizoma et Radix Polygoni Cuspidati	[74]
	ε-Viniferins	Xinjiang wine grapes (*Vitis Vinifera*)	[95]
**CE**	*Trans*-resveratrol *Cis*-resveratrol	White wines Red wines	[96]

## Data Availability

The data presented in this study are available in the article.

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
