# Peer review of "Resveratrol: Its Path from Isolation to Therapeutic Action in Eye Diseases"

_antioxidants, 2022, doi:10.3390/antiox11122447_

Round 1

Reviewer 1 Report

The authors conducted a comprehensive review of resveratrol focusing on eye diseases. The topic of the manuscript is of relevance and general interest to the readers.  Overall, the manuscript is very well written and the authors performed a thorough, detailed, and focused review of the literature.  The only suggestion is to add an objective statement in the Introduction section.  This was not clear.

Author Response

Dear Reviewer,

Thank you very much for your comments and suggestions.

Please see my answer below:

  1. The only suggestion is to add an objective statement in the Introduction section. This was not clear.

In accordance with the request expressed by the reviewer, a clear and concise objective statement was formulated and added at the beginning of the introduction section of the manuscript.

Reviewer 2 Report

Dear Authors,

the manuscript is comprehensive and presents an extensive study on resveratrol biochemistry, biological properties and delivery systems.

Please, add the antioxidant activity of resveratrol in ocular diseases in the paragraph 8 (line 512). The reference paper to cite is: 

Lançon A, Frazzi R, Latruffe N. Anti-oxidant, anti-inflammatory and anti-angiogenic properties of resveratrol in ocular diseases. Molecules, 2016 doi.org/10.3390/molecules21030304

Author Response

Dear Reviewer,

Thank you very much for your comments and suggestions.

Please see my answer below:

  1. Please, add the antioxidant activity of resveratrol in ocular diseases in the paragraph 8 (line 512).

According to the reviewer's suggestion, paragraph 8 has been updated with the indicated paper written by Lancon et al. (2016), as well as with two other papers related to the antioxidant activity of resveratrol measured in various cell types of the eye. The papers were also added in Table 2.

Reviewer 3 Report

The authors of the antioxidants-2038380 manuscript, entitled “ Resveratrol: its path from isolation to therapeutic action in eye 2 diseases” a very important review publication.

Resveratrol (3,5,4′-trans-trihydroxystilbene), which belongs to the stilbene family, is a polyphenolic phytoalexin. It is mainly found in grape skins and seeds but also in peanuts, berries, tea and It is a component of red wine. Resveratrol it exhibits a wide range of biological properties, among others, anti-glycation, antioxidant, anti-inflammatory, neuroprotective, and antineoplastic effects. However, due to its low bioavailability and quick metabolism, the possibility of utilizing its properties is limited, which makes RSV an ongoing subject of numerous clinical trials.

The authors discussed the biochemical pathways of resveratrol and its derivatives biosynthesis. They proposed figures on the connection between the resveratrol biosynthesis pathway and the biochemical pathway for synthesis of flavonoids.

In the next part, they referred to the methods of extraction, isolation and analysis of stilbenes/ supported by two tables. The bioavailability and safety of resveratrol and the anti-angiogenic modulatory effect of resveratrol are also presented.

Finally, the antioxidant role of resveratrol in eye diseases is discussed.

Thank you for such a well-prepared manuscript. In my opinion, this very well prepared article should be published in its present form.

Author Response

Dear Reviewer,

Thank you very much for your comments. We are delighted that this manuscript has been assessed as well prepared and ready for publication. In this light, we hope that the manuscript will be relevant for the scientific community and that the addressed topic will arouse the interest of the readers.

Thank you again for the careful review given to this work and for your valuable suggestions!